# Electronic Nicotine Delivery Systems Use and Periodontal Health—Findings from the Population Assessment of Tobacco and Health Study

**DOI:** 10.3390/healthcare12010025

**Published:** 2023-12-21

**Authors:** Amir Mohajeri, Martin S. Lipsky, Rachana Hegde, Jody Chiang, Man Hung

**Affiliations:** 1College of Dental Medicine, Roseman University of Health Sciences, South Jordan, UT 84095, USA; mlipsky@pdx.edu (M.S.L.); rhegde@roseman.edu (R.H.); jchiang698@student.roseman.edu (J.C.); mhung@roseman.edu (M.H.); 2Institute on Aging, Portland State University, Portland, OR 97207, USA; 3George E. Wahlen Department, Veterans Affairs Medical Center, Salt Lake City, UT 84148, USA; 4Huntsman Cancer Institute, Salt Lake City, UT 84112, USA; 5School of Medicine, University of Utah, Salt Lake City, UT 84113, USA

**Keywords:** ENDS use, periodontal disease, oral health, bone loss, PATH study

## Abstract

(1) Background: Electronic nicotine delivery systems (ENDSs) are rapidly increasing in the U.S., however, information about their long-term risks and benefits remains limited. This study examined the relationship between ENDS use and periodontal health among U.S. adults. (2) Methods: Data came from 33,822 adults who participated in the 2016–2018 wave of the Population Assessment of Tobacco and Health (PATH) study. Inclusion criteria were adults without a history of cigarette smoking or diabetes. Logistic regression analysis was performed to estimate the associations between ENDS use and a history of periodontal disease, with multivariable logistic regression adjusting for factors associated with poor oral health. (3) Results: Of the study participants, 2321 were never ENDS users, 38 were regular ENDS users, and 512 were non-regular ENDS users. Compared to never ENDS users, regular ENDS users had higher odds of poor periodontal health including bone loss around teeth. Regular ENDS use was also independently associated with higher odds of poor oral health compared to non-regular ENDS users. (4) Conclusions: This study suggests an association between ENDS use and increased risk of periodontal health issues in the United States. These findings align with previous research linking ENDS use to poor oral health.

## 1. Introduction

Despite the rapid increase in the use of electronic nicotine delivery systems (ENDSs) in the U.S. [1,2], there is scant information about their long-term health effects [3]. These systems, classified as non-combustible, battery-powered devices, provide nicotine, flavorings, and other additives to the user through an aerosol. ENDS devices marketed in the U.S. include personal vaporizers, vape pens, hookah pens, electronic cigarettes (e-cigarettes), e-hookahs, and e-pipes [4].

Periodontal disease is a common inflammatory condition affecting the supporting tissues of the tooth. It is strongly linked to overall health, making it one of the top 100 causes of disability-adjusted life years globally [5]. Moreover, untreated periodontitis is a leading cause of tooth loss in the United States [6]. Conventional cigarette smoking increases the risk of periodontal disease through a variety of mechanisms including impaired immunity, direct toxicity to fibroblasts and gingival tissue, alteration in tissue oxidation gradient, alteration of the microbiota, heightened inflammation, and an increased rate of alveolar bone loss [7,8]. Despite being perceived as safer and less harmful than regular cigarettes [9], e-cigarettes and other electronic nicotine products still pose health risks [10]. Some electronic nicotine products produce toxin-containing aerosols, and the urine of e-cigarette users contains harmful compounds such as tobacco-specific nitrosamines, polycyclic aromatic hydrocarbons, and heavy metals [1]. One in vitro study found that e-cigarette vapor caused inflammation of the gingival epithelium cells, similar to that observed in cells exposed to conventional cigarette smoke [11].

While few studies have specifically addressed the impact of ENDS use on oral health [4], most evidence suggests that ENDS use increases the risk of periodontal disease [12]. A systematic review showed that e-cigarette use can mitigate oral symptoms for conventional smokers, but harmful oral health sequelae still occur as a result of using e-cigarettes [12]. One recent study found that the e-cigarette aerosol affects the oral microbiome [12] and impairs immune function, making e-cigarette users more prone to infection than conventional cigarette smokers or non-smokers [13]. Another study reported that ENDS use increased the odds of being diagnosed with gum disease and alveolar bone loss [4]. A small pilot study found that tobacco smokers who switched from smoking conventional cigarettes to e-cigarettes experienced a significant increase in gingival inflammation [14]. In contrast, one study showed improved periodontal health in smokers who switched to e-cigarettes from conventional tobacco products [15], and in another small study, cigarette smokers experienced more periodontal inflammation compared to those who vaped or never smoked [3].

This study sought to explore the associations between ENDS use patterns and periodontal health using a nationally representative sample of adults in the United States. We hypothesized that ENDS use would be associated with increased odds of poor periodontal health after controlling for the use of other tobacco products and other known risk factors. These findings will add to the existing epidemiologic research about the impact of ENDS use on periodontal health such as bone loss and bleeding gums, and help policymakers and healthcare professionals understand the implications of ENDS products on health and patient care.

## 2. Materials and Methods

### 2.1. Data Source

This study analyzed data from Wave 4 of the Population Assessment of Tobacco and Health (PATH) survey collected between 1 December 2016 and 3 January 2018. The PATH survey is a national longitudinal cohort study of U.S. adults (years) and youths (12 to 17 years) that examines tobacco use and associated health effects. The PATH study represents a collaboration between the National Institute on Drug Abuse (Bethesda, MD, USA), the National Institutes of Health (Bethesda, MD, USA), the Center for Tobacco Products (Beltsville, MD, USA), and the Food and Drug Administration (FDA, Silver Spring, MD, USA). Scientists at the National Institutes of Health (NIH) and the Food and Drug Administration (FDA) lead the PATH study in partnership with Westat (Rockville, MD, USA). This Maryland based research organization has expertise in survey design, questionnaire development, data collection, and analysis [16]. The survey draws upon several sources including items about nicotine dependence based in part on the National Epidemiological Survey on Alcohol and Related Conditions Survey, physical health questions drawn from the National Health and Nutrition Examination Survey, and health-related items derived from validated screening tools such as the Global Appraisal of Individual Needs (GAIN) and Patient Reported Outcomes Measurement Information System [17]. PATH collects interview data through audio computer-assisted self-interviews (ACASI). Wave 1 began in 2011, and Wave 4 represents the third follow-up wave of the Wave 1 cohort. All Wave 1 respondents stayed eligible for the Wave 4 interview if they remained residents of the U.S. and were not incarcerated. Among the Wave 1 cohort members, 27,757 adult interviews were completed in Wave 4. Data were also collected from 6065 adults in the replenishment sample who were asked to participate in the PATH study for the first time. The study used a combined total of 33,822 adults completing Wave 4 interviews. More details of the PATH study methods, data collection, and sampling have been previously described [17].

### 2.2. Inclusion/Exclusion Criteria

The study group consisted of adults participating in the Wave 4 PATH survey. Participants who smoked or had diabetes were excluded because both conditions are associated with periodontal disease [18]. Diabetes induces periodontitis by causing an excessive inflammatory response to the periodontal microflora [19], and cigarette smoking is estimated to generate more than half of the periodontal disease cases in adults in the United States [20]. The analysis also excluded survey responses with missing values for diabetes, cigarette smoking, and ENDS use. Figure 1 displays a selection criteria flowchart.

### 2.3. Demographic Information

Sociodemographic variables included household income, the participant’s age at the time of the interview, race/ethnicity (Non-Hispanic white, Non-Hispanic black, Non-Hispanic other, and Hispanic), grade level, and sex. Age was divided into groups (18–24, 25–34, 35–44, 45–54, 55–64, 65 years old or older), education into levels (less than high school, GED, high school graduate, some college, and bachelor’s degree or advanced degree), and annual income (less than 10,000 USD, 10,000–24,999 USD, 25,000–49,999 USD, 50,000–99,999 USD, 100,000 USD or more).

### 2.4. ENDS Use

Interviewers conducted in-home interviews using an ACASI method. The survey included questions about the use of electronic nicotine products defined as e-cigarettes, vape pens, personal vaporizers and mods, e-cigars, e-pipes, e-hookahs, and hookah pens [21]. Adults were asked if they had ever used an electronic nicotine product one or two times (yes vs. no; defined in this study as ever ENDS users) or regularly every day or some days (yes vs. no; defined in this study as regular ENDS users). A composite variable consisting of the ever-use of ENDS products and the never-use of ENDS products regularly was defined in this study as non-regular ENDS users. For analysis, the ENDS status variable was dummy coded, with 0 assigned to participants who answered no to “having ever used any ENDS product”, 1 to those who reported yes to using electronic nicotine products regularly every day or some days, and 2 to participants who answered yes to “having ever used any ENDS product” and no to using electronic nicotine products regularly every day or some days.

### 2.5. Outcome Measures

All health outcomes in the PATH study were self-reported and included gum disease, bone loss around teeth, gum bleeding, loose teeth, and tooth extraction. The presence of gum disease, bone loss, bleeding, and loose teeth were determined by the following PATH survey questions: Have you ever been told by a dentist, hygienist, or other health professional that you have gum disease? Have you ever been told by a dentist, hygienist, or other health professional that you lost bone around your teeth? Have you ever observed any bleeding after brushing or flossing, or due to other conditions in your mouth? Have you ever had any teeth become loose on their own, without an injury? In all four items, the response options were yes or no, with 1 indicating yes and 0 meaning no. Tooth loss was assessed by having those who responded yes to tooth loss, answering the question: How many of your permanent teeth have been removed because of tooth decay or gum disease? The tooth loss variable was converted to a binary variable where “≥1” was assigned to participants who reported having at least one permanent tooth removed because of tooth decay or gum disease and “<1” for those who had never had a permanent tooth extracted in their lifetime. Appendix A details the parts of the survey used for this study.

### 2.6. Covariates

PATH collected data on several risk factors associated with poor oral health. For our analysis, we included the following risk factors: alcohol use, other tobacco use, dental visit history, flossing, and perceptions about the health risks of smoking. The other tobacco use variable was subcategorized as cigar use, pipe use, hookah use, snus use, smokeless tobacco use, and dissolvable tobacco use. Participants also reported their beliefs about the health risks of smoking using a 5-point scale where lower numbers indicate more agreement. For analysis purposes, the responses were collapsed into a three-point scale, Agree, Disagree, and Neither agree nor disagree, by grouping the strongly agree and agree responses under Agree and strongly disagree and disagree grouped as Disagree. Based on the American Academy of Periodontology’s recommendation to include daily flossing as a part of a regular oral hygiene routine, open-ended numeric responses to flossing were categorized into two groups: one less than seven times/week (less than daily) and the other seven or more times/week (once daily or more) [22]. Appendix A displays the questions related to all of these variables.

### 2.7. Statistical Analysis

The distribution of the participant’s demographic characteristics, oral health outcomes, and covariates according to ENDS use patterns were calculated. Chi-square tests compared the frequencies of nominal variables by ENDS use status (Table 1). We also tested the association of poor oral health across covariates to select predictors for multivariable logistic regression models. Using multivariable logistic regression analysis, we calculated the adjusted odds ratios (OR) and 95% confidence intervals (CI) for the association between ENDS use and poor oral health after controlling for the confounding factors identified as significantly different (see Table 2). An assessment of overfitting risk was conducted using cross-validation. A ratio of 80% was used for training, while 20% was used for model evaluation. All analyses were conducted using IBM SPSS Statistics V27, with significance set as *p* < 0.05.

## 3. Results

Of the 33,822 participants who completed the Wave 4 survey, 6576 were missing data on cigarette smoking or diabetes status, and 24,330 had a history of cigarette smoking or diabetes and were excluded from the analysis. An additional 23 participants were missing data on ENDS use status, yielding a final sample of 2893 respondents for analysis (see the flowchart in Figure 1).

Between 2016 and 2018, an estimated 80.2% of non-cigarette smoking and non-diabetes adults in the U.S. were never ENDS users, 1.3% were regular ENDS users, and 17.7% were non-regular ENDS users (Table 1). Approximately 3% of the study population reported having gum disease, 2% had bone loss around teeth, 41% had gum bleeding, 7% reported having a tooth become loose without an injury, and 10% reported having at least one missing tooth (see Table 1).

### 3.1. Main Outcome Measures

Table 2 shows the bivariate analysis of bone loss and tooth extraction. Prevalence varied significantly by ENDS use status for bone loss around teeth and tooth loss, with 5.4% of regular ENDS users responding yes to questions about bone and tooth loss. Only 2.1% of never ENDS users and 0.6% of non-regular ENDS users reported bone loss around teeth. Additionally, 10.6% of never ENDS users, 7.3% of non-regular ENDS users, and 5.3% of regular ENDS users reported having had at least one permanent tooth removed (Table 2).

After adjusting for the factors identified in Table 2 as being significantly associated with bone loss around teeth and tooth loss, regular ENDS users were almost five times more likely than never ENDS users to experience bone loss (OR = 4.82, 95% CI = 1.04 to 22.35) (see Table 3). Furthermore, ENDS use showed a dose–response for bone loss, with the likelihood of bone loss being almost 13 times greater in regular ENDS users (OR = 12.2, 95% CI = 1.94 to 76.37) compared to never-ENDS users (Table 4) and more than twice the risk of the non-regular user. The likelihood of having tooth loss did not differ significantly between regular ENDS users and never ENDS users or non-regular ENDS users. To assess overfitting in our models, we performed cross-validation. For 20% and 80% of the sample, we found that the correlations between the outcomes and predicted models were small and insignificant, indicating a low risk of overfitting.

### 3.2. Secondary Outcomes

Significant differences were observed among regular, non-regular, and never ENDS users with ENDS more likely to be younger (age < 24 years) (*p* < 0.001), non-Hispanic white (*p* < 0.001), less educated (*p* < 0.001), and reporting more alcohol use (*p* < 0.001) (Table 1). The vast majority of all respondents reported receiving some dental treatment, with regular ENDS users (97.4%) reporting the highest percentage, followed by non-regular ENDS users (93.9%) and never ENDS users (91%). Regular ENDS users (7.9%) reported the lowest prevalence of daily flossing, while 11.3% of non-regular ENDS users and 15.5% of never-ENDS users reported flossing daily (Table 1). Participants aged 65 years old and older (10.4%) and those who did not believe smoking caused mouth cancer (15.6%) were significantly more likely to experience bone loss (Table 2). Bone loss around teeth was also significantly more common among those with a history of other tobacco use (2.5%). People with other non-Hispanic backgrounds, a bachelor’s or advanced degree, people with an annual income of 10,000 to 24,999 USD, without an alcohol abuse history, and those who visited the dentist and had a higher prevalence of bone loss, but these differences did not achieve statistical significance. Table 2 also shows the outcome of the bivariate analysis for tooth loss. Tooth loss was significantly more common among participants aged 65 years old and older (50.8%), non-Hispanic black (14.8%), participants with less education (GED) (23.3%), those earning between 25,000 and 49,999 USD per year (12.6%), those with adequate oral hygiene practices (14.4%) (flossing once a day or more), those who did not visit the dentist (15.5%), and those who did not believe smoking could cause mouth cancer (34.4%). A higher prevalence of tooth loss was observed among men, those without alcohol abuse histories, and those with other tobacco histories, but these differences were not significant (Table 2).

## 4. Discussion

Poor oral health is a significant health concern because it influences the person’s health and quality of life [23]. There was a 40% increase in the number of people with untreated oral disease between 1990 and 2015, affecting an estimated 3.5 billion individuals worldwide [24]. This increase in disease burden has sparked the interest of policymakers, and oral health is now being recognized as a controllable priority for global public health improvement.

Although ENDS products are thought to be a safer alternative to traditional cigarette smoking [25], the statistical data show an exponential rise in ENDS use among young adults, and nicotine content in the majority of ENDS products can have negative consequences for oral health [13,26]. However, due to the lack of randomized controlled trials, a causal connection between ENDS use and detrimental effects on oral health cannot be firmly established [25]. In addition, surprisingly few studies have characterized ENDS-related oral health outcomes [25]. This study examined the influence of different ENDS use patterns on periodontal diseases in the United States adult population.

When placed alongside other emerging evidence from epidemiological and experimental studies, the study findings suggest that ENDS use is a risk indicator for periodontal disease in the United States [3,27].

Our findings for regular ENDS users compared to non-regular ENDS users are particularly interesting. Those who used ENDS products regularly every day or some days were almost 13 times more likely than non-regular ENDS users to have bone loss. However, the odds of bone loss for non-regular ENDS users were statistically indistinguishable from never ENDS users. Similarly, Atuegwu and colleagues [4] found the odds of having gum disease, bone loss, and any periodontal disease were 1.76, 1.67, and 1.58, respectively, for longitudinal ENDS users compared to never ENDS users. However, they also found no significant association between longitudinal and non-longitudinal ENDS users.

Several of our findings related to ENDS use parallel other studies. A significantly higher percentage of regular ENDS users was found in the age group of 18–24 years. Others also found that this age group was more likely to vape [28] for reasons such as the attraction of flavored products, the option of varying nicotine content, and peer usage [29]. Our finding that males are more likely to use ENDS products is consistent with other studies that also reported a higher prevalence of ENDS use in men [30]. More than half of regular ENDS users had a lower education level (high school), a result comparable to a study that found higher levels of education decreased the odds of using e-cigarettes [31]. One reason may be that those with less education are less aware of the harmful effects of electronic tobacco products [32]. Given the adverse effects of ENDS use, it is disturbing that fewer regular ENDS users flossed more than seven times per week than non-users. This suggests that promoting good oral health offers an opportunity to mitigate the potential harms of ENDS use.

The association between ENDS use and poor oral health in this study is generally consistent with the findings of other studies of ENDS use and periodontal disease [4,14,15,33]. Although these studies used a wide range of risk factors, diverse study populations, and different study designs, most have been limited by few participants. Furthermore, even though diabetes and cigarette smoking are major risk factors for periodontitis, few studies have considered the health effects of ENDS use regarding these factors. Controlling for these confounding variables helps establish ENDS use as an independent risk factor. This is unsurprising since several plausible pathophysiologic mechanisms support the connection between ENDS use and periodontal disease. Nicotine affects gingival blood flow, cytokine production, immune cell function, connective tissue turnover and is likely a key mechanism [34]. Other possible mechanisms include aerosolized toxins, carcinogens, and chemical substances such as flavorings that can trigger irritation, inflammation, and damage to the oral tissue [35].

## 5. Limitations

The main limitation is the self-reported nature of periodontal status and ENDS product use, which has the potential for recall bias and misreporting. For example, the participants may interpret the phrase “Regular use of electronic nicotine products” differently. Ideally, quantifiable numbers such as weekly electronic nicotine product usage might have yielded more accurate answers. Another limitation is that the survey might not have included all potentially confounding variables. For example, despite recent evidence that physical activity may contribute to periodontitis [36], that data were not included in our model. Another limitation is that the PATH survey was conducted in the United States and may not be representative of other countries. However, there is no reason to suspect that individuals from other nations would have different biological responses. Finally, the study was limited by data regarding when each patient started and stopped using ENDSs, which prevents finding a chronological correlation between the onset of bone loss and the start of ENDS use.

## 6. Conclusions

In this nationally representative sample of U.S. adults, ENDS use was significantly associated with increased odds of poor oral health, even after controlling for known confounders. These results suggest that regular ENDS use may be a risk indicator of developing periodontal disease including bone loss around teeth. Given the rapid growth in the use of ENDS products and the observed associations with oral health, medical professionals should inquire about and document the ENDS products used at every patient visit.

## Figures and Tables

**Figure 1 healthcare-12-00025-f001:**
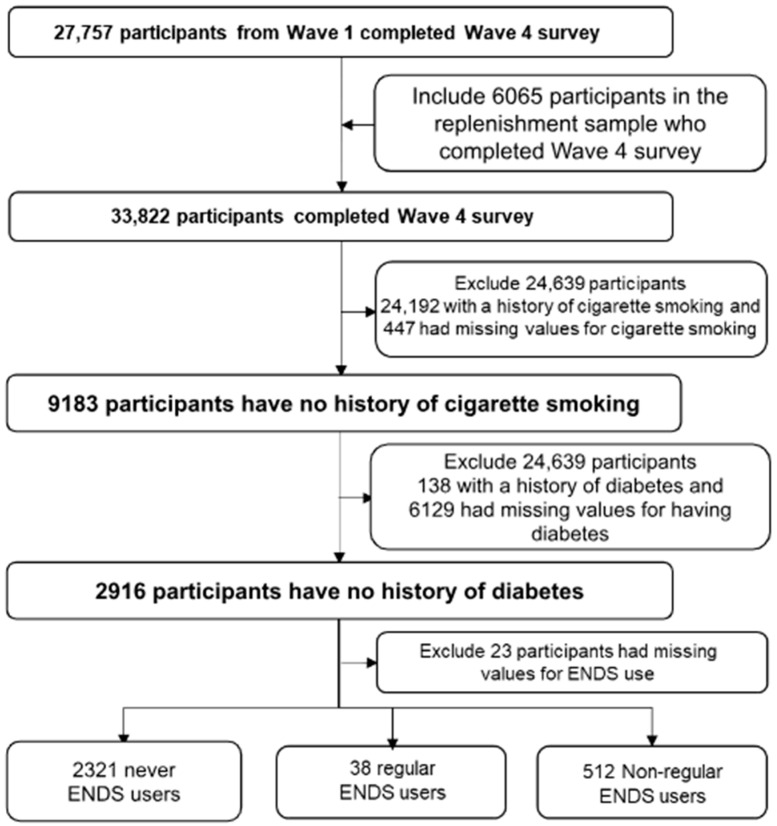
Selection criteria flowchart.

**Table 1 healthcare-12-00025-t001:** Demographic and other characteristics for the total sample of Wave 4 non-cigarette smokers and non-diabetes adults and by ENDS use status.

	Total	Never ENDS User *n* = 2321	Regular ENDS User *n* = 38	Non-Regular ENDS User *n* = 512	*p*-Value ^1^
	*n* (Percent)	*n* (Percent)	*n* (Percent)	*n* (Percent)	
Age group					
18 to 24 years old	2428 (84.6%)	1909 (82.3%)	36 (94.7%)	483 (94.3%)	<0.001 *
25 to 34 years old	144 (5%)	123 (5.3%)	1 (2.65%)	20 (3.9%)	
35 to 44 years old	105 (3.7%)	99 (4.3%)	1 (2.65%)	5 (1%)	
45 to 54 years old	70 (2.4%)	68 (2.9%)	0 (0%)	2 (0.4%)	
55 to 64 years old	55 (1.9%)	54 (2.3%)	0 (0%)	1 (0.2%)	
65 or more years old	67 (2.3%)	66 (2.8%)	0 (0%)	1 (0.2%)	
Gender					
Female	1489 (51.9%)	1224 (52.7%)	10 (26.3%)	255 (49.8%)	0.003 *
Male	1379 (48.1%)	1094 (47.1%)	28 (73.7%)	257 (50.2%)	
Race/Ethnicity					
Non-Hispanic White	1251 (44.2%)	991 (43.4%)	24 (63.2%)	236 (46.5%)	0.031 *
Non-Hispanic Black	518 (18.3%)	441 (19.3%)	5 (13.2%)	72 (14.2%)	
Non-Hispanic Other	286 (10.1%)	234 (10.2%)	3 (7.9%)	49 (9.6%)	
Hispanic	774 (27.4%)	617 (27%)	6 (15.8%)	151 (29.7%)	
Education					
Less than high school	532 (18.7%)	439 (19.1%)	7 (18.9%)	86 (16.9%)	<0.001 *
GED	73 (2.6%)	62 (2.7%)	1 (2.7%)	10 (2%)	
High school graduate	1081 (38%)	844 (36.7%)	19 (51.4%)	218 (42.7%)	
Some college	813 (28.5%)	640 (27.8%)	10 (27%)	163 (32%)	
Bachelor’s degree or advanced degree	349 (12.3%)	316 (13.7%)	0 (0%)	33 (6.5%)	
Annual income					
Less than 10,000 USD	515 (20.6%)	407 (20.2%)	7 (21.2%)	101 (22.2%)	0.125
10,000 to 24,999 USD	482 (19.3%)	401 (19.9%)	2 (6.1%)	79 (17.4%)	
25,000 to 49,999 USD	502 (20.1%)	403 (20%)	7 (21.2%)	92 (20.3%)	
50,000 to 99,999 USD	531 (21.2%)	441 (21.9%)	7 (21.2%)	83 (18.3%)	
100,000 or more USD	470 (18.8%)	361 (17.9%)	10 (30.3%)	99 (21.8%)	
Alcohol history					
No	616 (36%)	568 (39.4%)	6 (35.3%)	42 (16.7%)	<0.001 *
Yes	1093 (64%)	873 (60.6%)	11 (64.7%)	209 (83.3%)	
Other tobacco use					
No	2184 (76.1%)	1933 (83.3%)	10 (26.3%)	241 (47.1%)	<0.001 *
Yes	687 (23.9%)	388 (16.7%)	28 (73.7%)	271 (52.9%)	
Behaviors and attitudes					
Visit to the dentist	2625 (91.6%)	2107 (91%)	37 (97.4%)	481 (93.9%)	0.039 *
Interdental cleaning, ≥7 times/wk (behavior)	420 (14.7%)	359 (15.5%)	3 (7.9%)	58 (11.3%)	0.026 *
Mouth cancer in smokers (Belief)	2729 (95.1%)	2207 (95.1%)	37 (97.4%)	485 (94.7%)	0.476
Outcome Measures					
Ever gum disease	85 (3%)	71 (3.1%)	2 (5.3%)	12 (2.3%)	0.482
Ever bone loss around teeth	54 (1.9%)	49 (2.1%)	2 (5.4%)	3 (0.6%)	0.020 *
Ever gum bleeding	1167 (40.7%)	923 (39.8%)	15 (39.5%)	229 (44.7%)	0.120
Ever loose teeth	187 (6.5%)	157 (6.8%)	1 (2.6%)	29 (5.7%)	0.407
Ever tooth extraction	283 (9.9%)	244 (10.6%)	2 (5.3%)	37 (7.3%)	0.048 *

^1^ *p*-value derived from the chi-square test for the comparison of variables among regular and never ENDS users. * Difference is significant.

**Table 2 healthcare-12-00025-t002:** Association of risk factors with bone loss around teeth and tooth loss.

Characteristics	Categories	Having Bone Loss		Having Tooth Extractions	
		%	*p*-Value	%	*p*-Value
Age (years)	18 to 24 years old	1.4	<0.001 *	6	<0.001 *
	25 to 34 years old	2.1		19.7	
	35 to 44 years old	2.9		31.4	
	45 to 54 years old	2.9		37.7	
	55 to 64 years old	7.3		38.2	
	65 or more years old	10.4		50.8	
Gender	Female	1.9	0.986	10.5	0.314
	Male	1.9		9.3	
Race/Ethnicity	Non-Hispanic White	1.8	0.682	7.6	<0.001 *
	Non-Hispanic Black	1.7		14.8	
	Non-Hispanic Other	2.8		10.9	
	Hispanic	1.8		9.1	
Education	Less than high school	1.5	0.258	11.2	<0.001 *
	GED	1.4		23.3	
	High school graduate	1.7		7.7	
	Some college	1.8		9.2	
	Bachelor’s degree or advanced degree	3.4		12.9	
Annual income	Less than 10,000 USD	1.4	0.401	11.9	0.004 *
	10,000 to 24,999 USD	2.7		11.5	
	25,000 to 49,999 USD	2.4		12.6	
	50,000 to 99,999 USD	1.9		7.4	
	100,000 or more USD	1.3		7	
History of alcohol use	No	2.9	0.494	14.6	0.147
	Yes	2.4		12.2	
Other tobacco use	No	2	0.034 *	9.9	0.976
	Yes	2.5		10	
Visit to the dentist (treatment)	No	1.2	0.465	15.5	0.002 *
	Yes	1.9		9.3	
Interdental cleaning, ≥7 times/wk (behavior)	No	1.8	0.417	9.2	0.001 *
	Yes	2.4		14.4	
Mouth cancer in smokers (Belief)	Disagree	15.6	<0.001 *	34.4	<0.001 *
	Agree	1.6		9.5	
	Neither agree nor disagree	4.6		14	
ENDS status	Never	2.1	0.020 *	10.6	0.048 *
	Regular	5.4		5.3	
	Non-regular	0.6		7.3	

* Difference is significant.

**Table 3 healthcare-12-00025-t003:** Multivariable logistic regression of risk factors associated with bone loss around teeth and tooth loss.

Characteristics	Categories	Having Bone Loss		Having Tooth Extractions	
		OR (95% CI)	*p*-Value	OR (95% CI)	*p*-Value
Age (years)	18 to 24 years old	Reference		Reference	
	25 to 34 years old	1.64 (0.48–5.61)	0.43	4.34 (2.53–7.43) *	<0.001
	35 to 44 years old	1.30 (0.37–4.62)	0.68	9.26 (5.42–15.84) *	<0.001
	45 to 54 years old	1.81 (0.42–7.85)	0.43	10.28 (5.41–19.55) *	<0.001
	55 to 64 years old	4.79 (1.59–14.47) *	0.005	12.71 (6.64–24.30) *	<0.001
	65 or more years old	7.10 (2.96–17.05) *	<0.001	17.38 (9.29–32.52) *	<0.001
Gender	Female	N/A		N/A	
	Male				
Race/Ethnicity	Non-Hispanic White	N/A		0.86 (0.58–1.28)	0.449
	Non-Hispanic Black			1.32 (0.87–1.99)	0.186
	Non-Hispanic Other			1.53 (0.91–2.57)	0.108
	Hispanic			Reference	
Education	Less than high school	N/A		Reference	
	GED			1.65 (0.79–3.45)	0.182
	High school graduate			0.75 (0.49–1.15)	0.180
	Some college			0.84 (0.54–1.31)	0.437
	Bachelor’s degree or advanced degree			0.55 (0.32–0.96) *	0.034
Annual income	Less than 10,000 USD	N/A		Reference	
	10,000 to 24,999 USD			0.73 (0.47–1.13)	0.156
	25,000 to 49,999 USD			0.89 (0.58–1.37)	0.607
	50,000 to 99,999 USD			0.42 (0.26–0.68) *	<0.001
	100,000 or more USD			0.43 (0.25–0.73) *	0.002
History of alcohol use	No	N/A		N/A	
	Yes				
Other tobacco use	No	Reference	0.38	N/A	
	Yes	0.71 (0.33–1.53)			
Visit to the dentist (treatment)	No	N/A		Reference	0.345
	Yes			0.79 (0.49–1.29)	
Interdental cleaning, ≥7 times/wk (behavior)	No	N/A		Reference	0.298
	Yes			1.22 (0.84–1.78)	
Mouth cancer in smokers (Belief)	Disagree	Reference		Reference	
	Agree	0.09 (0.03–0.26) *	<0.001	0.39 (0.15–1.01)	0.053
	Neither agree nor disagree	0.26 (0.07–1.04)	0.056	0.45 (0.14–1.43)	0.176
ENDS status	Never	Reference		Reference	
	Regular	4.82 (1.04–22.35) *	0.045	0.89 (0.19–4.07)	0.89
	Non-regular	0.39 (0.117–1.34)	0.135	1.12 (0.74–1.69)	0.58

N/A: Variables were not included in the multivariate models because they were non-significant in the bivariate analyses. * Statistically significant results.

**Table 4 healthcare-12-00025-t004:** The association between ENDS status and poor oral health considering non-regular ENDS users as a reference group.

Characteristics	Categories	Having Bone Loss		Having Tooth Extractions	
		OR (95% CI)	*p*-Value	OR (95% CI)	*p*-Value
ENDS status	Never	2.53 (0.75–8.52)	0.135	0.89 (0.59–1.35)	0.58
	Regular	12.2 (1.94–76.37) *	0.008	0.80 (0.17–3.75)	0.777
	Non-regular	Reference		Reference	

* Statistically significant results.

## Data Availability

The data for this study are freely available at https://www.icpsr.umich.edu/web/NAHDAP/studies/36498/datadocumentation (accessed on 13 December 2023).

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
