# Peer review of "Electronic Nicotine Delivery Systems Use and Periodontal Health—Findings from the Population Assessment of Tobacco and Health Study"

_healthcare, 2023, doi:10.3390/healthcare12010025_

Round 1
Reviewer 1 Report
Comments and Suggestions for Authors
A very interesting manuscript analyzing ENDS use and periodontal status and PATH study findings.
I have few points for elaboration.
1. Regarding periodontal bone loss, was there any clinical or radiographic examination to ascertain the bone loss or is it just by self-appraised responses for interviews?
2. As there are four categories of the subsets of population analyzed. It is important to know about the prevalence of racial background and its relationship with oral health, too loss, periodontal diseases.
3. It is difficult to establish causal relationship of ENDS with bone loss in the absence of clinical examination of cohorts, though the confounding factors have been considered in the data analysis.
4. There should be further details of all ENDS products used by the studied population.
5. The association between ENDS products with Oral health is interesting and common observation among smokers generally in clinical and epidemiological studies. The causal relationship and generalizability of the said products need cautious correlation. Though the results are quite convincing and valuable for oral health and medical care providers.
Reviewer 2 Report
Comments and Suggestions for Authors
The authors are exploring a timely topic with a clear methodology. I would advise some points for improvements.
- Abstract
I am not a big fan of the term gingival disease. Please, refer to the terminology approved by the 2017 Chicago Workshop
Abstract: the use of the term 'Risk factor' has to be sustained by a true prospective study design and by providing relative risk estimates or hazard ratios. Significant odd ratios refer to 'Risk indicators'.
- Introduction
Please, improve clarity of this passage: While ENDs use is thought to be less harmful than conventional cigarette smoking, it may still be harmful to health and can cause gingival inflammation similar to that seen in cells exposed to conventional cigarette smoking
In general, many sentences lack from proper referencing, for example: "While few studies address the impact of ENDS use on oral health, most evidence suggests that ENDS increase the risk of periodontal disease."
In the aim and in other passages, the authors used the term poor oral health, that is a too generic term. I would recommend being more specific.
- Methods and results
The paragraph 2.3 is void. I would recommend eliminating it and renaming the following: 'Demographics information'. Since ENDs use is the exposure and gum health the outcome, I would change the order of paragraphs 2.5 and 2.6.
The major limitation of the study clearly lays on the self-reported nature of the exposure, but most importantly of the outcome assessment. Indeed, how did the authors selected the questionnaire? Was it validated? It would be important to include the way in which it was converted in numerical variables in the manuscript.
The authors stated that significant factors in univariate analysis were selected for multivariate analysis. However, the risk of overfitting of their final model is high. How did the authors control for this risk?
Some results reported in Table 2 lack explanation. Do the authors think that not finding a 'dose-response' effect of ENDs use could be due to a big unbalance in the size of the different groups? Please, be also careful of the fact that the ones reported are not correlations.
- Discussion and conclusion
The discussion section is somehow missing an in-depth explanation of the findings. Interpretation in the context of the available literature could be particularly implemented, see for example doi: 10.1177/00220345221086272 or doi: 10.1007/s00784-023-05162-4. The same is valid for some biological rationale explaining the detrimental effects of ENDs.
Also here, many statements lack referencing.
The statement 'Our finding that non-regular ENDS users were less likely than regular ENDS users to have periodontal disease is novel and has not been previously reported' is too strong.
Among other known confounders, physical activity has been recently associated to periodontitis (doi: 10.1111/jcpe.13766). If it not possible to adjust for this factor, discuss at least in the limitation.
Please, make clear in the limitation that 'a major risk of information bias may exist in relation to the self-reported nature of the assessment periodontal status and the consequent risk of misclassification'.
Comments on the Quality of English Language
Some minor English language improvements are required throughout the manuscript, but the overall quality is good.
Reviewer 3 Report
Comments and Suggestions for Authors
Dear authors!
Indeed, the number of people using electronic nicotine delivery systems is growing every year all over the world, that is why it is important to understand the long-term health effects of such systems. The presented article, aimed at identifying the effect of electronic nicotine delivery systems on the condition of the oral cavity, is very interesting, and its results are of practical importance, because they will help to predict or prevent undesirable manifestations of these systems.
The design of the study is simple and well understood, and also corresponds to the purpose of the study. The manuscript has a clear structure and is easy to read. The "Materials and methods" section is described in sufficient detail to ensure the reproducibility of the research and complies with the necessary requirements. The results of the study are presented in detail both in the text and in tables. The conclusions are formulated correctly and correspond to the aim of the study. Literary references are relevant. Excessive number of self-citations is excluded.
However, there are some moments that are worth paying attention to:
1. There is a certain file format in the requirements for manuscripts on the journal's website. It is worth noting that the template posted on the publisher's website contains line numbering, which greatly facilitates understanding and communication between authors and reviewers (https://www.mdpi.com/journal/jpm/instructions#:~:text=Accepted%20File%20Formats,for%20further%20details).);
2. There should be no abbreviations in the title. Decipher the terms "ENDS" and "PATH";
3. In the "Materials and Methods" section it is worth adding information about the inclusion criteria in addition to information about the exclusion criteria;
4. There is no information in the paragraph 2.3. "Measures", it should be deleted;
In general, the article leaves a good impression, thus, I can recommend this article for publication.
Reviewer 4 Report
Comments and Suggestions for Authors
The authors analyzed the PATH survey and found that regular ENDS users are more likely to carry bone loss in their medical history. The introduction is informative and clearly explains the aim of the study. The authors should remove the numbers between parentheses from the abstract. In addition, the authors should add the number of the study participants (512 ENDS regular users, 38 non-regular users, and 2321 controls). The introduction is informative and provides an essential background to the manuscript. Anyway, the authors should add the aim of the study and list their hypotheses. In addition, the authors should displace the last two sentences of the introduction to the discussion. The methods are well-written, and this reviewer has no remarks on such a section. The results section obliges the reader to study the tables to understand the manuscript. Therefore, this reviewer suggests splitting the results into main and secondary outcomes to improve the readability of such a section. In the discussion section, the authors state that “adults who have not visited the dentist will be classified as not having dental issues.” Can the authors disambiguate such a sentence? In addition, a limitation of the study is the lack of data regarding the start and the last of the use of ENDS for each patient; this is a limitation because it prevents finding a chronological coherence between the starting point of the ENDS use and the onset of the bone loss.
In my opinion, the study should be considered for publication after MAJOR REVISIONS.
With warm regards
Comments on the Quality of English LanguageBefore publication, the manuscript needs English editing.
Round 2
Reviewer 2 Report
Comments and Suggestions for Authors
I have no further comments
Author Response
Thank you very much for taking the time to help us improve the manuscript. We greatly appreciate it.
Reviewer 4 Report
Comments and Suggestions for Authors
The authors replied to my concerns and modified the manuscript according to most of my suggestions. Anyway, the manuscript needs professional English editing before publication.
In my opinion, the study should be considered for publication after MINOR REVISIONS.
With warm regards
Comments on the Quality of English LanguageThis reviewer suggests English professional editing for the current manuscript.
Author Response
We sought professional English editing services to enhance the quality of our manuscript. Please see the attached revision. Thank you very much for your suggestion.